# Spheroid Culture System Methods and Applications for Mesenchymal Stem Cells

**DOI:** 10.3390/cells8121620

**Published:** 2019-12-12

**Authors:** Na-Eun Ryu, Soo-Hong Lee, Hansoo Park

**Affiliations:** 1Department of Integrative Engineering, Chung-Ang University, Seoul 06974, Korea; ryuskdms234@gmail.com; 2Department of Medical Biotechnology, Dongguk University, Seoul 06974, Korea

**Keywords:** 3D cell culture, spheroid culture, biomaterials

## Abstract

Owing to the importance of stem cell culture systems in clinical applications, researchers have extensively studied them to optimize the culture conditions and increase efficiency of cell culture. A spheroid culture system provides a similar physicochemical environment in vivo by facilitating cell–cell and cell–matrix interaction to overcome the limitations of traditional monolayer cell culture. In suspension culture, aggregates of adjacent cells form a spheroid shape having wide utility in tumor and cancer research, therapeutic transplantation, drug screening, and clinical study, as well as organic culture. There are various spheroid culture methods such as hanging drop, gel embedding, magnetic levitation, and spinner culture. Lately, efforts are being made to apply the spheroid culture system to the study of drug delivery platforms and co-cultures, and to regulate differentiation and pluripotency. To study spheroid cell culture, various kinds of biomaterials are used as building forms of hydrogel, film, particle, and bead, depending upon the requirement. However, spheroid cell culture system has limitations such as hypoxia and necrosis in the spheroid core. In addition, studies should focus on methods to dissociate cells from spheroid into single cells.

## 1. Introduction

Stem cells are valuable resources in regenerative medicine with clinical and research applications (Table 1). Particularly, human mesenchymal stem cells have secretory properties constituted by anti-inflammation, angiogenesis, and immune reaction regulation factors [1,2]. The primary characteristic of stem cells is stemness, represented by their ability of self-renewal, which generate new same cells from the original stem cells, and multipotency, which allows production of new differentiated cells having relatively limited potential [3]. Another characteristic of stem cells is clonality, which is related to lineage of stem cells [4]. Stemness enables the stem cells to have potential for application in numerous biological therapeutic tools such as cell-based therapy, high-throughput pharmacology, drug screening, and tissue engineering. However, stemness is actually maintained in the in vivo microenvironment, which provides growth factors in addition to the cell–cell or cell– extracellular matrix interactions. Therefore, conditions of the cell culture are very essential to facilitate the properties of stemness, maintenance, and proliferation. To fully exploit the properties of stem cells, development of cell culture methods that can increase the proliferation of cells in terms of cell count and superior quality are important. Previously, studies have attempted to develop methods for enhancing stemness and proliferation during in vitro cell culture process [5,6].

Cells constantly require biological signals from the substrates in cellular niches. These signals encourage proliferation and enhance cellular viability. However, the signals can also inhibit proliferation and biological activation of cells, thereby checking the growth [7]. In general, traditional two-dimension cell culture systems, wherein the cells grow as a monolayer, face some limitations in the realization of in vivo multi-cellular conditions [8]. These limitations restrict the cellular studies involving multi-cellular features and cancer cells [9]. When stem cells grow in two-dimensional cultures, maintenance of the differentiation potential and stemness is relatively more difficult than in stem cells growing in actual multi-cellular conditions [6]. However, three-dimensional cell culture systems can reconstitute conditions similar to that in an in vivo microenvironment (Table 2). Three dimensional systems construct cell–cell and cell–extracellular matrix (ECM) interaction networks (Table 2), which play a significant role in various cellular mechanisms, subsequently maintaining the cellular properties [10]. Researchers can attempt to decrease the limitations of conventional monolayer culturing and develop progressive cellular study methods (Table 2).

Spheroid culture system is a promising three-dimensional cell culture method. Morphology of stem cells cultured in spheroid culture system is different from that of cells in the monolayer culture system. In addition, mesenchymal stem cells of spheroids maintain their intrinsic phenotypic properties by cell–extracellular matrix interactions (Table 2) [11].

Spheroid stem cell cultures promote the expression of transcriptional factors of stemness markers such as Oct-4 and Nanog (Table 2). Spheroid stem cells secrete higher levels of cytokines and chemokines that affect the proliferation, viability, and migration of cells and also secrete higher angiogenesis than those secreted in the monolayer stem cells (Table 2). Spheroid stem cells regulated by hypoxia-induced upregulation of gene expression have properties of apoptosis resistance, improved viability, and secretion of angiogenic factors and chemokines (Table 2) [12,13].

## 2. Mechanism of Spheroid Formation

In a cell culture suspension, cells tend to aggregate and go through the process of self-assembly. Self-assembly means single cells constitute multi-cellular spheroids by themselves. Self-assembly is a natural phenomenon that happens during embryogenesis, morphogenesis, and organogenesis. It is affected by various factors, including gradients of nutrients, oxygen, and growth factors in cell culture medium, as well as cellular paracrine factors. Cell culture medium permeates inside the spheroids by diffusion. The gradient of diffusion is induced by increasing the spheroid size during spheroid culture. The bigger the size attained by the spheroids, the harder it becomes for the medium to reach the core of the spheroids. In addition, the rate of production and consumption of factors can affect self-assembly [19].

Adhesion and differentiation of cells affect the formation of multi-cellular spheroids. In particular, cadherin and integrin are directly related to the mechanism of spheroid formation. The process of spheroid formation is divided into several steps. Firstly, single cells present within the suspension agglomerate to form loosely adhesive cell spheroids. In this step, extracellular matrix fibers including complementary binding of peripheral cell surface to integrin encourages preliminary aggregation. Next, E-cadherin promotes strong adhesion of initial cell aggregate by creating homophilic binding between cadherins of peripheral cells. In addition, β-catenin complex facilitates cellular signal transduction. Actin can also affect agglomeration and stemness by promoting contacts between adjacent cells [20]. As a result, strong adhesive multi-cellular spheroids are formed [21].

E-cadherin (CDH1), a Ca^2+^-dependent homophilic transmembrane adhesion molecule, could be a central component of spheroid formation [22,23]. The effects of E-cadherin have been demonstrated in cellular experiments involving human breast cancer cell lines and mouse embryonic stem cells [24]. For example, in a mouse embryonic stem cell study, E-cadherin-mediated cell attachment initiates embryonic body agglomeration. In addition, if the reaction of E-cadherin on the embryonic body with α-mouse E-cadherin antibodies is blocked, there is considerable inhibition of embryonic body agglomeration [25]. Integrins are transmembrane adhesion proteins composed of α-subunits and β-subunits of heterodimers that facilitate the cell-ECM connection during cell invasion and migration. Apart from E-cadherin, β_1_-integrin also plays a role in the attachment of the early spheroid formation. Interaction of integrin-ECM affects multi-cellular spheroid formation rates [21].

## 3. Spheroid Formation Methods

### 3.1. Technical Methods

#### 3.1.1. Pellet Culture

In this system, cells are concentrated to the bottom of the tube by centrifugal force. Cell–cell adhesions are maximized by proximity of the single cells at the bottom of the tube (Figure 1a), (Table 3). To harvest the cell pellet, supernatants are removed, and cell pellets are resuspended in spheroid formation cell culture medium. After estimating the cell count, cells in medium are dispensed into each well of a 96-well U bottom plate with cell repellent surface [19,26].

Pellet culture can be used to induce differentiation of mesenchymal stem cells. In particular, a pellet culture system is suitable for stem cell differentiation by chondrogenesis since the interaction between adjacent cells in a pellet culture microenvironment is similar to the interaction in pre-cartilage condensation occurring during embryonic development. In pellet culture, mesenchymal stem cells can change their morphological shape from fibroblastic to polygonal in a manner similar to that in chondrocytes. Therefore, pellet culture system can be used for the study of signal pathways of chondrogenesis and for assessing the chondrogenic potentiality of stem cells [27,28].

#### 3.1.2. Liquid Overlay

Liquid overlay culture technique, also called the static suspension culture, forms spheroids by interrupting the adhesion of cells on non-adherent culture plates (Figure 1b), (Table 3). A non-adherent culture layer is typically composed of agar or agarose gel. Agarose is a very efficient material for the inhibition of cell attachment and is superior to agar with respect to its non-adherent properties. Since the cell attachments are inhibited, cells spontaneously form spheroids above the non-adherent surface by promoting cell–cell adhesive molecules [19,29,30].

Despite excellent non-adherent properties of agarose, this biomaterial has drawbacks in terms of culturing cancer cells. Agarose has trouble in interacting with tumor cells and is unable to activate the specific signaling pathways related to reaction of tumor cells to therapy process. Recently, hyaluronic acid can be a suitable alternative biomaterial that can replace agarose. It has the capability to interact with surface receptors of cancer cells during cancer progression. This interaction enhances transduction of cellular signals related to proliferation, angiogenesis, survival, and differentiation, as well as resistance to therapeutics [31,32].

#### 3.1.3. Hanging Drop

Hanging drop culture technique allows single cells to aggregate and fabricate spheroids in the form of droplets (Figure 1c), (Table 3). By controlling the volume of the drop or density of cell suspension, it is possible to control the spheroid size [33]. The novel hanging drop array platform is capable of efficiently forming definite size spheroids [34]. This technique can form circular spheroids having a narrow distribution of size with 10% to 15% variation coefficient, while the spheroid growth in non-adherent surface culture methods has 40% to 60% variation coefficient [35]. A general method involves starting from a monolayer cell culture, after which the cells are prepared as suspension and diluted with culture medium to attain the desired cell density. Subsequently, the cell suspension is dispensed into wells of a mini-tray with the help of a compatible multistep or multichannel pipette. A lid is placed on the mini-tray and the entire mini-tray is reversed upside down. The cell suspension drops attached on the mini-tray would stay on the reversed surface by surface tension. In this method, spheroids are formed as droplets owing to simultaneous action of surface tension and gravitational force [19,36].

Besides the adjustable size of a spheroid, hanging drop system has other advantages. There is no requirement of expensive or professional equipment to form spheroids for small scale experiments. A huge quantity of spheroids can be produced readily by multichannel pipetting and can be harvested by scraping lids of culture dishes [33]. In addition, mesenchymal stem cells cultured via hanging drop system can secrete considerable quantities of potent anti-inflammatory as well as anti-tumorigenic factors [37].

#### 3.1.4. Spinner Culture

Spinner culture technique refers to the technique wherein the cell suspension in spinner flask bioreactor containers is continuously mixed by stirring (Figure 1d), (Table 3). The resultant spheroid is dependent on size of the bioreactor container [9,38]. Conditions of the fluid and mass in the containers are affected by the convectional force of the stirring bar, which is crucial to form the spheroid. A high stirring rate induces damage to the spheroid cells. However, an extremely slow rate of stirring allows spheroid cells to sink to the bottom of the container, resulting in inhibition of spheroid formation in the container [19].

In addition to adipogenesis, osteogenic differentiation of mesenchymal stem cells is also boosted by improved expression of osteogenic markers such as osteopontin and osteocalcin in the spinner system [39].

#### 3.1.5. Rotating Wall Vessel

Rotating wall vessel reconstructs microgravity by constant circular rotation [40]. Due to constant rotation, cells are continuously in a suspended state in the vessel (Figure 1e), (Table 3) [41]. This microgravity can affect gene expression of mesenchymal stem cells. In microgravity conditions, chondrogenic and osteogenic gene expression of stem cells reduces, whereas adipogenic gene expression is elevated [42]. This is because microgravity inhibits expression of *Collagen I* of the osteoblastic marker gene and integrin/Collagen I signaling pathway during the osteoblastic differentiation [43]. In addition, microgravity suppresses stress fiber development and improves intracellular lipid accumulation. However, reduction of osteogenic gene expression by microgravity can be regulated. Expression of RhoA protein switches these microgravitational effects and improves expression of the markers of osteoblastic differentiation of mesenchymal stem cells [44]. Expression of chondrogenic genes is increased by regulation of the p38 MAPK activation pathways [45].

#### 3.1.6. Microfluidics

This microfluidic culture technique, also called lab-on-a-chip technique, is used for applications such as single cell analysis, genetic assays, and drug toxicity studies. This culture method has microscale dimensions corresponding to the scale of in vivo microstructures (Figure 1f), (Table 3). In addition, microfluidic devices easily enable microscale control of the environment, mimicking the in vivo three-dimensional environment. One of the features of the microfluidic method is that it integrates multiple processes including cell capture, mixing, detection, and cell culturing. Another feature is a considerably high cell throughput for cell analysis. Microfluidic devices employ materials permeable to oxygen and growth factors affecting proliferation. This characteristic feature of microfluidics technology can decrease hypoxia, which is an unavoidable disadvantage of spheroid culture [46].

Recently developed fluidic systems overcome the limitations posed by the conventional fluidic system and offer advantages such as diversity of design and cost reduction through smaller requirements for specimens and reagents for cell transport assays [47]. Presently, the fluidic system can produce a distinct concentration of analyte mixtures and facilitates real-time monitoring of living cells. In addition, this system can optimize cell culture conditions for the proliferation and differentiation of stem cells, and be used for tissue engineering processes such as organ replacement and tissue regeneration, and in future clinical trials [48,49,50]. The currently used microfluidics system can be used to develop a co-culturing system related to the generation of microvascular network using mesenchymal stem cells. The co-culture system can also induce formation of a human microvascular network [51].

#### 3.1.7. Magnetic Levitation 

Magnetic levitation-based culturing makes use of magnetic particles and integration with hydrogels according to the given conditions. In the magnetic levitation system, cells are mixed with magnetic particles and subjected to magnetic force during cell culture (Figure 1g), (Table 3). This system utilizes negative magnetophoresis, which can imitate a weightlessness condition, because positive magnetophoresis can hinder the attainment of weightlessness [52]. Due to magnetic force, the cells incorporated with magnetic particles stay levitated against gravity. This condition induces the geometry change of cell mass and promotes contact between cells, leading to cell aggregation. In addition, this system can facilitate multi-cellular co-culturing with agglomeration of different cell types [53,54].

When mesenchymal stem cells and magnetic particles are cultured with collagen gel, particle internalization takes place. Spheroid formation can be reproducible and reduces necrosis in the spheroid core, thus maintaining its stemness as a spheroid [54]. However, some groups have demonstrated that artificially manipulated gravity can lead to changes in cellular structures and can result in apoptosis [55,56]. 

### 3.2. Using Biomaterials Methods

#### 3.2.1. Hydrogels

Hydrogels are widely used for cell culture studies. Hydrogels have been fabricated using biocompatible materials such as alginate [57,58], fibrin [59,60], collagen [54] and hyaluronic acid [61,62]. The primary properties of hydrogels is that mesenchymal stem cells can be entrapped in them (Figure 2a), (Table 4). This method effectively improves the viability of cells while reducing cellular apoptosis. Furthermore, osteogenic differentiation potential is stably maintained and secretion of proangiogenic factors is activated in the hydrogel-entrapped cells compared to that in the non-entrapped cells of the monolayer culture [11,57,59]. Activated secretion of proangiogenic factors implies increased angiogenic potential and highly correlates to improved osteogenesis [63,64].

Physicochemical biomimetic properties of hydrogels similar to those of the extracellular matrix are capable of offering functional niches promoting the self-renewal potential and wound healing. These properties of hydrogels improve angiogenetic capacity and stemness of the cells (Table 4) [65]. By adjusting the physical properties of hydrogel materials, the size of a spheroid can be optimized. To control the size of spheroids, weak adhesive materials can be used and physically embossed patterns on the surface of the hydrogel should be fabricated [66]. In addition, another main property of a hydrogel is the capability to deliver cells directly. Hydrogels can also be prepared in an injectable form that can directly deliver stem cells to in vivo models and compensate for the necrotic or defective tissues [65,67].

Hydrogels have been developed to study the microenvironment of cancer cells. The stiffness of hydrogels can affect the phenotype and growth of cancer cells. The number and size of cancer cells tend to reduce when cultured in stiff hydrogels. However, tumorigenicity of cancer cells increases after in vivo transfer in softer hydrogels. This stiffness of hydrogels can be optimized by modifying the concentrations of the composing materials [68].

#### 3.2.2. Biofilms

Films made of biomaterials can be constructed by various methods such as photolithography and stamping (Figure 2b). Tumor cell spheroids can be cultured on films for their role in cancer drug discovery [69,70]. Apart from tumor cells, stem cells can be cultured on films. Stemness marker expression and differentiation potential are increased during culturing on film (Table 4) [71,72]. Adhesion and proliferation of stem cells can be enhanced by changing the composition and concentration of the film materials (Table 4) [73]. The component ratios used in the films are critical to the size of spheroids and rate of spheroid fabrication, as well as cellular adhesion and proliferation. In a previous study, hyaluronic acid(HA) modified chitosan film was found to form larger spheroids and induce cell aggregation in lesser time than the unmodified chitosan film [74]. The size of spheroids is also affected by the thickness of the film. Reducing thickness of the film leads to decreased spheroid size [75].

One of the biomaterials used in such films is chitosan. Culturing on chitosan films can improve angiogenesis, chemotaxis, and self-renewal [13,76]. Recently, graphene has been investigated as a cell culturing material [77,78]. Graphene films can provide distinctive environments beneficial to neurogenesis. Moreover, the neurons differentiated on graphene films have a remarkably keen sense of external stimulations. Graphene is believed to be capable of adjusting neural differentiation and growth of mesenchymal stem cells [79].

#### 3.2.3. Particles

Particulate factors have been used in spheroid cultures to control the cell culture microenvironment (Figure 2c). A drawback of spheroid cultures is the inadequate supply of nutrients and oxygen to the core of the spheroid. This is accounted by a rise in the diffusion gradient with increased spheroid size. However, particles within spheroids are capable of controlling conditions inside the spheroids during culturing. Consequently, the viability and proliferation of cells improve (Table 4) [80,81].

Particles are capable of regulating stem cell differentiation by controlling the extracellular environment [82]. Differentiation is also regulated by encapsulating stem cells in these particles. A previous study demonstrated that mesenchymal stem cell encapsulating particles, including a nanofibrous meshwork, could induce osteogenic differentiation [83]. However, particles can inhibit specific stem cell differentiation while inducing differentiation of other stem cells. This is achieved by controlling mechanotransductional mechanisms. Particles act as obstacles of internal adhesion between adjacent cells of spheroids. Alteration of mechanical force, including internal adhesion, surface tension, and interfacial tension in a spheroid, leads to biased differentiation of the stem cells [84].

The desired delivery of growth factors using particles can modulate the spheroid microenvironment. By transferring suitable growth factors into spheroids, differentiation can be spatially controlled [85,86].

## 4. Applications of Spheroid

### 4.1. Study of Tumors

Tumor cells are affected by cellular structures and extracellular matrix. The conventional 2D culture system has limitations pertaining to tumor cell culture. Spheroid culture system is a promising method for the study of tissue structure, signaling pathways, and immune activation of cancer cells.

Single tumor cells may form multi-cellular tumor spheroids mixed with other types of cells in a non-adherent 3D culture system, which is more effective in creating cellular heterogeneity [87]. Morphologies of these tumor spheroids are affected by spheroid culture microenvironment. In accordance with the conditions of culture microenvironment, morphologies may form aggregated circles, entangled bundles, elongated ovals, or star-shaped spheroids.

Spheroid form is the most suitable model for cancer study because it has a limited oxygen concentration at its core. This hypoxic nature of spheroids is the primary advantage of spheroid culture. However, in the case of tumors, spheroids larger than 500 μm in diameter undergo necrosis at their core [87] and have a concentration gradient of biological factors similar to tumor cells due to restricted diffusion of nutrients, oxygen, and growth factors [88].

### 4.2. Drug Screening

A study using an animal model has a limitation in disease modelling [89]. Presently, parameters of drug screening studies using a mouse model can possibly be overcome by adopting spheroid cell culture [90]. However, lack of uniformity in diameter or morphology of spheroids appears as new parameters for reproducible drug screening. By increasing the uniformity during the spheroid culture period, tumor spheroids can provide precise information on the diseases and suppress undesired side effects of the drugs under development [90,91].

In the context of tumor cell culture for drug screening, co-culture of normal cell and tumor cells can be a potential technique for reconstruction of the heterogenous multi-cellular environment for solid tumors as well as for promoting migration in tumors. The co-culture enables the investigation of interactions between tumor cells and peripheral multi-cellular environments. In addition, because normal host cells proximal to tumor cells can influence drug sensitivity of tumor cells, spheroid co-cultures can be used for drug screening study.

### 4.3. Regenerative Medicine

Transplantation is one of the most promising strategies for regenerative therapy. The currently used transplantation therapy has some drawbacks. In case of autografts, the amount of cellular supply is limited and the process of cell collection is cumbersome for the donor. However, allograft transplantation results in problems such as infection, inflammation, and host rejection [16]. Besides, injection in the form of single cells results in the limited immobility of injected cells at the site of the defect [92]. Injectable spheroids of stem cells are considered to improve the engraftment efficiency after transplantation [93]. After the implantation of spheroids, stem cells may be induced to differentiate into suitable cells for reconstructing the defective site [16,94]. Differentiation potential of spheroids has been demonstrated in vitro. Spheroid culture method improves differentiation potential compared to monolayer culturing [95].

Genetically modified spheroids have been developed for cell transplantation therapy [17,96]. These spheroids are prepared in the form of injectable suspensions. After transplanting these spheroids, the altered gene expression is maintained for a longer period of time in host tissues, whereas expression of cells cultured from monolayer plates decreases soon after transplantation. Thus, desired properties of cells transplanted in the host tissues are preserved by the process of spheroid culture [97].

## 5. Conclusions

Stem cells have shown applicability in various fields such as regenerative medicine as well as tumor and cancer research. Three-dimensional cultures enhance the applicability of stem cells by increasing the efficiency of culture. Spheroid culture system is an attractive method to overcome limitations of traditional monolayer culture. This system can resolve problems of monolayer culture such as the limited realization of in vivo multi-cellular microenvironments and it can reconstruct biological signal pathways of cell–cell and cell–ECM interactions, which encourage proliferation and viability of cells. Therefore, maintenance of the differentiation potential, stemness and intrinsic phenotypic properties is improved. To conclude, development of spheroid culture is essential to further optimize formation of spheroids and utilize them as resource in the medical field.

## Figures and Tables

**Figure 1 cells-08-01620-f001:**
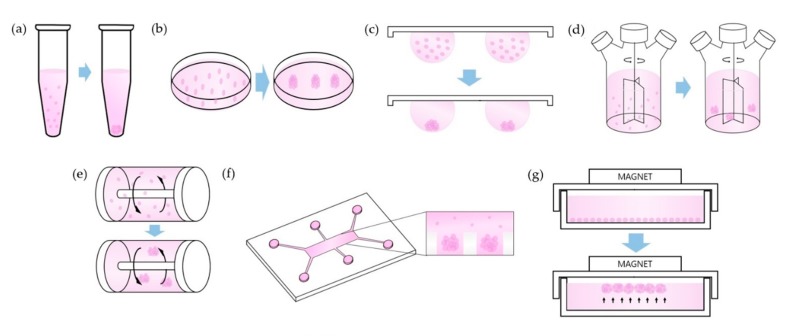
Schemes of technical methods. (**a**) Pellet Culture, (**b**) Liquid Overlay, (**c**) Hanging Drop, (**d**) Spinner Culture, (**e**) Rotating Wall Vessel, (**f**) Microfluidics, (**g**) Magnetic Levitation.

**Figure 2 cells-08-01620-f002:**
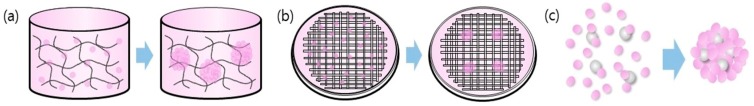
Schemes of using biomaterials methods. (**a**) Hydrogels, (**b**) biofilms, (**c**) particles.

**Table 1 cells-08-01620-t001:** Applications of spheroid of Mesenchymal stem cells.

Applications of Spheroid	Examples
**Tumor model**	MSCs have resistance to chemotherapies and produce biochemical responses similar to parental tumors. Therefore MSCs-based modeling is usable to predict in vivo therapeutic efficacy [9].
**Biology research**	Co-culture of MSCs with other cells has been used to analyze cell–cell interaction. The cell–cell interactions are crucial to function of tissues [9].
**Tissue Engineering**	MSCs have been used for organ reconstruction. Spheroid of MSCs provide advantageous conditions for organ reconstruction. The MSCs can be transplanted to patient [9].
**Transplantation therapy**	Differentiation of MSCs is enhanced by spheroid culture system. In vivo tissue (cartilage [14,15], bone [16], spinal cord [17], nerve [18]) formation may be enhanced after MSCs spheroid transplantation.

**Table 2 cells-08-01620-t002:** Advantages and disadvantages of spheroid culture.

Advantages	Disadvantages
• facilitate cell–cell and cell–matrix interaction • provide a similar physicochemical environment to the in vivo • maintain intrinsic phenotypic properties • promote the stemness marker expression • secrete cytokines, chemokines and angiogenic factors • improve viability and proliferation	• have diffusion gradient with increased spheroid size and lack of nutrients in the core of spheroid

**Table 3 cells-08-01620-t003:** Properties of technical methods.

Technical Method	Properties
Pellet Culture	use centrifugal force to concentrate cells
Liquid Overlay	use non-adhesive materials to inhibit cell attachment
Hanging Drop	use surface tension and gravitational force
Spinner Culture	use convectional force by stirring bar
Rotating Wall Vessel	use constant circular rotation of vessel
Microfluidics	use microfluid flow and materials permeable to soluble factors
Magnetic Levitation	use magnetic force to levitate cells

**Table 4 cells-08-01620-t004:** Properties of biomaterials.

Biomaterial	Properties
**Hydrogel**	• entrap cells during culture and can deliver cells as injectable form. • provide an environment similar to extracellular matrix and improve viability, stemness and angiogenetic capacity of stem cells.
**Biofilms**	• increase stemness, differentiation potential, adhesion and proliferation of stem cells.
**Particles**	• control mechanotransductional mechanisms inside the spheroid and improve viability and proliferation.

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
