# Peer review of "Spheroid Culture System Methods and Applications for Mesenchymal Stem Cells"

_cells, 2019, doi:10.3390/cells8121620_

Round 1
Reviewer 1 Report
The review paper by Ryu et al. described the overview of spheroid cultures for mesenchymal stem cells (MSC), including mechanistic aspect, methods and applications. The paper mainly deals with MSC spheroids but includes spheroids using other cell-types (e.g. tumor cells), possibly causing confusion. The point should be improved prior to publication. The manuscript should be carefully checked for spelling and grammar.
L13: physsicochemical -> physicochemical
L14: tranditional -> traditional
L15: suspention -> suspension
L21: ared -> are, fom -> form
L23: dessociated -> dissociate
L28: medicin -> medicine
L74: cultur -> culture
L116: reconstruct -> reconstructs
L200: Leviatation -> Levitation
L305: appear -> appears
L337: a -> an
L343: reource -> resource
Author Response
Response to Reviewer 1 Comments
Reviewer 1: The review paper by Ryu et al. described the overview of spheroid cultures for mesenchymal stem cells (MSC), including mechanistic aspect, methods and applications.
Point 1. The paper mainly deals with MSC spheroids but includes spheroids using other cell-types (e.g. tumor cells), possibly causing confusion. The point should be improved prior to publication.
Response 1: We agree with the reviewer’s comments and changed the line 91. In section 3.1.2, We just wanted to explan additional information about the material used to form spheroid.(line 130, 131)
Point 2: The manuscript should be carefully checked for spelling and grammar.
L13: physsicochemical -> physicochemical
L14: tranditional -> traditional
L15: suspention -> suspension
L21: ared -> are, fom -> form
L23: dessociated -> dissociate
L28: medicin -> medicine
L74: cultur -> culture
L116: reconstruct -> reconstructs
L200: Leviatation -> Levitation
L305: appear -> appears
L337: a -> an
L343: reource -> resource
Response 2: We thank the reviewer for these positive comments and changed every word that reviewer suggested. (line 13, 15, 21, 24, 28, 78, 170, 204, 315, 347, 353)
Reviewer 2 Report
In this article, the author organized the different methods and applications of spheroid culture system, it is a promising issue in regenerative medicine and drug screening, but there are some content could be improved by addressing the following points:
There are some typographical errors on page 1, line 21(form), line 28(medicine); page 2 line 71(is affected); page 9 line 337(an attractive method), line 343(resource). It is recommended to provide a comparison table for the organization of spheroid properties, including formation methods, advantages, disadvantages or the spheroid size, features of the spheroid, it would be benefited to read and understand that the differences between those spheroid culture systems. Mesenchymal stem cells have essential functions in clinical applications. In the references cited by this article, the application of mesenchymal stem cell spheroids includes tissue regeneration (cartilage, bone and nerve) and cell transplantation therapy. It is recommended to provide a table for the application of mesenchymal stem cell spheroid. It would highlight the point of this article. Many materials fit the definition of hydrogels, both naturally occurring and synthetic. And the properties of these materials would not exactly similar. In section 3.2.1, the author only mentions the broad term “hydrogel” and does not elaborate on the effect of the type of hydrogel on the cells.Author Response
Response to Reviewer 2 Comments
Reviewer 2: In this article, the author organized the different methods and applications of spheroid culture system, it is a promising issue in regenerative medicine and drug screening, but there are some content could be improved by addressing the following points:
Point 1: There are some typographical errors on page 1, line 21(form), line 28(medicine); page 2 line 71(is affected); page 9 line 337(an attractive method), line 343(resource).
Response 1: We thank the reviewer for these positive comments and corrected every word that reviewer suggested. (line 21, 28, 74, 347, 353)
Point 2: It is recommended to provide a comparison table for the organization of spheroid properties, including formation methods, advantages, disadvantages or the spheroid size, features of the spheroid, it would be benefited to read and understand that the differences between those spheroid culture systems.
Reponse 2: We agree with the reviewer’s comments and added tables (Table 2, 3, 4)
Point 3: Mesenchymal stem cells have essential functions in clinical applications. In the references cited by this article, the application of mesenchymal stem cell spheroids includes tissue regeneration (cartilage, bone and nerve) and cell transplantation therapy. It is recommended to provide a table for the application of mesenchymal stem cell spheroid. It would highlight the point of this article.
Response 3: We agree with the reviewer’s comments and added table 1.
Point 4: Many materials fit the definition of hydrogels, both naturally occurring and synthetic. And the properties of these materials would not exactly similar. In section 3.2.1, the author only mentions the broad term “hydrogel” and does not elaborate on the effect of the type of hydrogel on the cells.
Response 4: We changed and added some setences in 3.2.1 to emphasize the properties of hydrogel. (line 229. 242). Additionally we summarized properties of hydrogel in table 4.